# Effects of Electric-Toothbrush Vibrations on the Expression of Collagen and Non-Collagen Proteins through the Focal Adhesion Kinase Signaling Pathway in Gingival Fibroblasts

**DOI:** 10.3390/biom12060771

**Published:** 2022-06-01

**Authors:** Kumiko Nakai, Hideki Tanaka, Kyoko Fukuzawa, Jyunya Nakajima, Manami Ozaki, Nobue Kato, Takayuki Kawato

**Affiliations:** 1Department of Oral Health Sciences, Nihon University School of Dentistry, Tokyo 101-8310, Japan; tanaka.hideki@nihon-u.ac.jp (H.T.); kyoko.fukuzawa@gmail.com (K.F.); ozaki.manami@nihon-u.ac.jp (M.O.); deno21006@g.nihon-u.ac.jp (N.K.); kawato.takayuki@nihon-u.ac.jp (T.K.); 2Division of Functional Morphology, Dental Research Center, Nihon University School of Dentistry, Tokyo 101-8310, Japan; 3Department of Oral and Maxillofacial Surgery, Nihon University School of Dentistry, Tokyo 101-8310, Japan; deju20015@g.nihon-u.ac.jp

**Keywords:** electric-toothbrush vibrations, gingival fibroblasts, extracellular matrix, collagen, fibronectin, elastin, focal adhesion kinase signaling pathway

## Abstract

Electric-toothbrush vibrations, which remove plaque, are transmitted to the gingival connective tissue via epithelial cells. Physical energy affects cell function; however, the effects of electric-toothbrush vibrations on gingival extracellular matrix (ECM) protein expression remain unknown. We aimed to examine the effects of these vibrations on the expression of ECM proteins—type I collagen (col I), type III collagen (col III), elastin, and fibronectin (FN)—using human gingival fibroblasts (HGnFs). HGnFs were seeded for 5 days in a six-well plate with a hydrophilic surface, exposed to electric-toothbrush vibrations, and cultured for 7 days. Subsequently, the mRNA and protein levels of col I, col III, elastin, and FN were examined. To investigate the role of focal adhesion kinase (FAK) signaling on ECM protein expression in vibration-stimulated cells, the cells were treated with siRNA against protein tyrosine kinase (PTK). Electric-toothbrush vibrations increased col I, col III, elastin, and FN expression; promoted collagen and non-collagen protein production; and enhanced FAK phosphorylation in HGnFs. Moreover, PTK2 siRNA completely blocked the effects of these vibrations on the expression of col I, col III and elastin mRNA. The results suggest that electric-toothbrush vibrations increase collagen, elastin, and FN production through the FAK-signaling pathway in fibroblasts.

## 1. Introduction

The deterioration of oral health is a major public healthcare concern [1]. Periodontal diseases are usually caused by an imbalance between host defense and environmental factors such as smoking, poor nutrition, and a high percentage of periodontopathogenic bacteria. Promoting regular oral hygiene can contribute to the maintenance of a functional dentition throughout life [2].

The electric toothbrush was conceived in the 1950s with an aim to improve and facilitate oral hygiene. It was designed to target patients with limited motor skills as well as orthodontic patients who have difficulties in keeping their teeth hygienic and healthy. Geriatric patients with impaired manual skills can also benefit from the use of electric toothbrushes [3]. Since the 1980s, electric toothbrushes have rapidly developed to become an established alternative to manual toothbrushes [4,5]. The electric toothbrush has been widely used for daily oral self-care, and its rate of use is increasing.

The currently available electric toothbrushes have various mechanisms of action, such as oscillation–rotation, side-to-side sonic action, counter oscillation, circular motion, ultrasonic action, and ionic action. However, only two technologies are dominantly used: oscillation–rotation and sonic action. In the former, a small round brush head first rotates in one direction and then in the opposite direction, whereas in the latter, a traditional brush head laterally moves side to side at a high vibrational speed (mean frequency: 250 Hz, in other words, 30,000 brush strokes per minute) [6,7,8].

The effectiveness of electric toothbrushes has been investigated primarily in epidemiological studies, and continuous use of electric toothbrushes has been reported to reduce gingival inflammation and bleeding [9]. Ccahuana-Vasquez et al. [10] reported that the vibrations of electric toothbrushes significantly reduced plaque and gingival inflammation compared with those achieved with the use of manual toothbrushes. Similarly, other clinical studies have shown that electric toothbrushes are more effective than manual toothbrushes in plaque removal [11,12].

Fibroblasts in periodontal tissue play a crucial role in the remodeling of the extracellular matrix (ECM) by synthesizing and organizing connective tissue components. Fibroblasts, which make up 80% of the gingival tissue, produce ECM components (collagen, elastin, and laminin) and secrete growth factors, which help maintain periodontal tissue homeostasis, that is, host defense.

It has been reported that physical energy (e.g., ultrasound and electromagnetic fields) increases the ECM in bone and around junction cells [13,14,15]. These studies highlight the effect of physical energy on the ECM. The electric toothbrush can be used daily to physically stimulate the periodontium. The effect of electric-toothbrush vibrations on collagen production in connective tissue has been evaluated via in vivo studies with dogs and in vitro studies using human-derived cells [16,17]. Moreover, studies have been conducted to understand how the vibration affects the metabolism of hard tissues in the oral cavity [18,19,20]. Previous studies have reported accelerated tooth movement when high-frequency waves and mouthpieces are used together in orthodontic treatment [21,22]. Therefore, the vibrations from an electric toothbrush promote the remodeling of periodontal tissue, suggesting that the clinical use of this energy for bone and connective tissue repair may be useful. However, the mechanism involved in gingival ECM protein expression during the use of an electric toothbrush remains unknown.

In the present study, we stimulated human gingival fibroblasts (HGnFs) with an electric toothbrush and investigated the gene and protein expression of ECM proteins type 1 collagen (col I), col III, elastin, and fibronectin (FN). Experiments were conducted assuming that the vibrations of an electric toothbrush (Sonicare: HX8920B; Philips, Eindhoven, Netherlands) act on fibroblasts through the epithelial tissue. We also examined whether the signaling pathway and ECM protein expression was altered with the use of an electric toothbrush in HGnFs. To investigate the role of focal adhesion kinase (FAK) signaling in extracellular protein expression in cells stimulated with the vibrations, cells were treated with siRNA of protein tyrosine kinase (PTK) 2, a member of the FAK subfamily of PTKs.

## 2. Materials and Methods

### 2.1. Reagents

High-glucose (25 mM) Dulbecco’s modified Eagle’s medium (DMEM) and fetal bovine serum (FBS) were purchased from Gibco-BRL (Rockville, MD, USA). Fibroblast growth supplement (FGS) was purchased from ScienCell Research Laboratories (Carlsbad, CA, USA). Antibiotic solutions (penicillin/streptomycin) were purchased from Sigma-Aldrich (St. Louis, MO, USA).

The Sirius Red/Fast Green Collagen Staining Kit was purchased from Chondrex (Woodinville, WA, USA). The NucleoSpin RNA Kit, RNA PCR Kit (Prime Script), and TB Green^®^ Premix Ex Taq™ (Tli RNaseH Plus) solution were purchased from Takara Bio (Otsu, Japan). Enzyme-linked immunosorbent assay (ELISA; col1a1 and col3a1, FN, and elastin) kits were purchased from MyBioSource (San Diego, CA, USA).

Monoclonal IgG primary antibodies, including mouse anti-FAK, -phospho-FAK, and -GAPDH, were purchased from Santa Cruz Biotechnology (Dallas, TX, USA). Appropriate biotin-conjugated goat anti-mouse IgG secondary antibodies were purchased from Abcam (Cambridge, UK). Small interfering ON-TARGETplus Human PTK2 siRNA and Dharma FECT 1 Transfection Reagent were purchased from Dharmacon (Lafayette, CO, USA).

### 2.2. Stimulation with Electric Toothbrush

HGnFs (ScienCell Research Laboratories, Carlsbad, CA, USA) from human gingiva were seeded at a density of 4.7 × 10^4^ cells/well in a flexible bottom 6-well plate (BioFlex^®^ Culture Plates; Flexcell^®^ International Corp, Burlington, NC, USA) with a hydrophilic surface. Furthermore, they were cultured in high-glucose DMEM containing 2% FBS, 1% FGS, and 1% antibiotic solution (*v*/*v*) at 37 °C in 5% CO_2_.

Five days after seeding, the nylon bristles of an electric toothbrush (Gum Care brush) were set vertically against the outer surface of the flexible bottom of the plate, with a force of 15 g, by referring to Anadha et al. [23] (Appendix A). The cells on the inner surface of the flexible bottom were stimulated for 30 s by vibrations from the electric toothbrush. The duration of stimulation was determined based on the average time spent daily for oral cleaning using an electric toothbrush (30 s per quarter of the jaw). The cells were exposed to vibrations (value provided by the manufacturer: 261 Hz) once a day, and then cultured for 7 days.

To transmit vibrations in an effective manner, the bristles on the head were trimmed flat and set vertically against the bottom of the plate. A control was maintained with no vibration stimulation using an electric toothbrush.

### 2.3. Staining of Collagen and Non-Collagen Proteins

After washing with phosphate-buffered saline, the cells were soaked in Kahle’s fixative (ethanol, formaldehyde, and glacial acetic acid) and incubated for 10 min at 25 ± 1 °C. Subsequently, the fixed cells were soaked in the staining solution (Sirius Red/Fast Green Collagen Staining Kit, Chondrex, Woodinville, WA, USA) and incubated for 30 min. For a semi-quantitative analysis of the collagen and non-collagen proteins, the stain was eluted from the sample, and the OD values were measured at 540 and 605 nm using a spectrophotometer.

### 2.4. Real-Time Reverse Transcription (RT)-PCR

The total RNA was isolated from cells stimulated with/without vibrations using the NucleoSpin RNA Kit (Takara Bio, Otsu, Japan) according to the manufacturer’s instructions. After measuring the concentration of total RNA using a NanoDrop 1000 spectrophotometer (Thermo Fisher Scientific, Wilmington, DE, USA), cDNA was synthesized from 1 μg of RNA using the RNA PCR Kit (Takara Bio, Otsu, Japan). The resulting cDNA mixture was diluted 1:2 in sterile distilled water, and 2 µL of the diluted cDNA mixture was subjected to real-time RT-PCR with SYBR Green I dye. The reaction mixture comprised 14 μL of TB Green Premix Ex Taq solution containing 8 μM sense and antisense primers (Appendix A), and the reactions were conducted on a CFX Connect Real-Time PCR system (Bio Rad, Hercules, CA, USA).

The reactions involved 40 cycles at 95 °C for 5 s and 60 °C for 30 s. The reactions were performed in triplicate, and the specificity of the PCR products was verified using the instrument’s software based on the melting curve analysis. The 2^−ΔΔCt^ method was used to calculate gene expression relative to *GAPDH* expression.

### 2.5. ELISA

Culture supernatants were collected after stimulating the cells with/without vibrations for 7 days, and protein levels were measured using ELISA according to the manufacturer’s protocols. ELISAs were performed using the relevant kits for col I, col III, elastin, and FN to determine the levels of these proteins in the culture supernatants and cell lysate with extraction buffer containing 0.05% Triton X-100, 10 mM βmercaptoethanol, 0.5 mM phenylmethylsulfonyl fluoride, 0.5 mM ethylenediaminetetraacetic acid, and 25 mM Tris–HCl (pH 7.4). Each sample was analyzed in duplicate according to the manufacturer’s instructions. The concentration of each target protein was calculated based on the standard curves generated with the control peptide included in the ELISA kits.

### 2.6. Sodium Dodecyl Sulfate-Polyacrylamide Gel Electrophoresis (SDS-PAGE) and Western Blotting

The cells were lysed with extraction buffer containing 0.05% Triton X-100, 10 mM β-mercaptoethanol, 0.5 mM phenylmethylsulfonyl fluoride, 0.5 mM ethylenediaminetetraacetic acid, and 25 mM Tris-HCl (pH 7.4). Cell membranes were disrupted by sonication, and the samples were clarified by centrifugation. Supernatants containing 20 µg of intracellular proteins were mixed with 10 µL of sample buffer containing 1% SDS, 2 M urea, 15 mg/mL dithiothreitol, and bromophenol blue, and then heated at 95 °C for 5 min before loading. The proteins were resolved using 4–20% SDS-PAGE with a discontinuous Tris–glycine buffer system, transferred onto a polyvinylidene fluoride membrane using a semidry transfer apparatus, and probed with the indicated antibodies. Polyclonal or monoclonal IgG primary antibodies, including mouse anti-FAK, -phospho-FAK, and -GAPDH, were used with the appropriate biotin-conjugated goat anti-mouse IgG secondary antibodies. The membranes were labeled with streptavidin–horseradish peroxidase and visualized using a commercial chemiluminescence kit. To re-probe with different antibodies, the membrane was stripped at 25 ± 1 °C for 15 min with Restore PLUS Stripping Buffer.

### 2.7. Transfection with Small Interfering RNA

HGnFs were seeded at a density of 30 × 10^4^ cells/well in a 6-well plate. After 24 h, the cells were transfected with siRNA using the Dharma FECT 1 (Dharmacon, Lafayette, CO, USA) Transfection Reagent, as recommended by the manufacturer. The sequences of the siRNAs used are provided in Appendix A.

### 2.8. Statistical Analysis

The results are presented as mean ± standard deviations (SDs). Significant differences were determined using a one-way analysis of variance (ANOVA) followed by Tukey’s multiple comparison test. Differences with a *p*-value < 0.05 were considered statistically significant, and those with a *p*-value < 0.01 were considered highly significant.

## 3. Results

### 3.1. Collagenous and Non-Collagenous Proteins in the Cultured Cell Layers 

The cultured cell layers were stained with Sirus Red and Fast Green on day 5 of culture. Deposition of non-collagenous (green) proteins was observed in the layer of cells stimulated with vibration and in the layer of the unstimulated control cells. On the contrary, it was difficult to confirm collagenous (red) deposition in microscopic images (Figure 1a). However, quantitative analysis revealed that the amounts of collagenous and non-collagenous proteins were significantly higher in stimulated cells than in the unstimulated control (Figure 1b).

### 3.2. Expression of ECM Molecules in HGnFs

The mRNA and protein levels of col I, col III, elastin, and FN in the cells on days 3, 5, and 7 of culture were examined using real-time PCR (Figure 2) and ELISA (Figure 3), respectively. On days 3, 5, and 7 of culture, the mRNA levels of col I, col III, and elastin were significantly higher in the cells stimulated with the vibration of the electric toothbrush than in the unstimulated control cells (Figure 2a–i). The mRNA level of FN was also significantly higher after the stimulation (except on day 3 of culture) than in the unstimulated control (Figure 2j–l).

### 3.3. Protein Levels of col I, col III, Elastin, and FN 

The protein levels of col III and elastin increased after exposure to the vibratory stimulation using the electric toothbrush on days 3, 5, and 7 of culture (Figure 3d–i). The FN expression also increased after the vibratory stimulation on days 5 and 7 of culture (Figure 3j–k). However, the vibratory stimulation did not have marked effects on the protein expression of col I; there was no significant difference in the col I protein level in between cells stimulated with the vibration of an electric toothbrush and the unstimulated control cells, except on day 3 of culture (Figure 3a–c).

### 3.4. Expression of FAK and phosphorylated FAK in HGnFs

The levels of FAK and phosphorylated FAK in cells stimulated with the vibrations of an electric toothbrush were examined (Figure 4). The levels of FAK and phosphorylated FAK increased 15 min after stimulation and remained elevated for 60 min after the stimulation. The ratio of the levels of phosphorylated FAK/FAK also increased after the vibratory stimulation (Figure 4).

### 3.5. Effect of PTK2 siRNA on the Expression of FAK, col III, Elastin, and FN in HGnFs

The expression of FAK in HGnFs was examined 5 days after treatment with PTK2 siRNA. FAK expression was lower in cells treated with PTK2 siRNA than in untreated control cells and cells treated with scrambled siRNA (Figure 5a). Thus, PTK2 siRNA attenuated the expression of FAK. Moreover, PTK2 siRNA completely blocked the effect of electric-toothbrush vibrations on the expression of col I, col III and elastin mRNA (Figure 5b–d) but did not completely block the effects on the expression of FN (Figure 5e).

## 4. Discussion

In this study, we examined the effects of vibratory stimulation of HGnFs using an electric toothbrush. The intensity of collagen and non-collagen protein staining in the cell layer on day 5 of culture increased following vibratory stimulation with the electric toothbrush. Based on these findings, it is not clear whether the electric toothbrush affects the expression of ECM in fibroblasts; however, we believe that such a hypothesis warrants further investigation. Both gene and protein levels of col I, col III, elastin, and FN were found to be increased following the vibratory stimulation.

Healthy gingiva consists of, on average, 4% junctional epithelium, 27% oral gingival epithelium, and 69% connective tissue, which includes cellular infiltrate accounting for 3–6% of the gingival volume. Collagen fiber bundles, which account for 55–60% of the connective tissue volume, provide structural support to the extracellular space of connective tissues. Owing to its rigidity and resistance to stretching, collagen is the perfect matrix material for skin, tendons, bones, and ligaments. There are 28 types of collagen, depending on the differences in the structures they form, and types I through IV are the most common [24,25,26].

Normal gingival tissue contains col I and III at a ratio of 5:1, accounting for 99% of the collagen, and type IV, which is associated with basal lamina, accounts for the remaining collagen volume. Col I contains two identical α1 chains and a chemically different α2 chain. Col I is virtually insoluble, owing to crosslinks that provide the structural and mechanical stability for normal function. Col III also consists of three α1 chains and is more fibrillar and extensible than type I.

Elastin is an elastic fiber with a crosslinked structure and is widely present in connective tissues [27]. In gingival connective tissue, fibroblastic cells are the main source of elastin secretion into the ECM. Elastin has been estimated to account for approximately 5% of the total volume of normal gingival tissues. Both elastin and collagen are responsible for modulating important biological and mechanical properties by cross-linking [27,28,29]. Collagen and elastin provide a scaffold for periodontal tissue regeneration and are important ECM components [30]. In the present study, the expression of both collagen and elastin increased from day 3 of culture, owing to the vibratory stimulation. Therefore, it is possible that the vibrations of the electric toothbrush increased the stability of the ECM in gingival tissue via the upregulation of collagen and elastin production in fibroblasts.

FN, a type of glycoprotein ubiquitous in the ECM, is assembled into a fibrillar matrix in all tissues and in all stages of life. FN and col I fibers are frequently found together in tissues and have been found to be colocalized in the secretory pathway of fibroblasts [31]. Several studies have shown that collagen fibrils do not accumulate in the absence of the FN matrix. After binding to collagen, FN aligns collagen fibrils parallelly and assists in the deposition of collagen fibrils [32,33,34]. These findings suggest that the FN matrix is a critical scaffold for collagen assembly and plays an important role in the composition of ECM proteins.

In this study, the gene and protein levels of FN increased following vibratory stimulation with an electric toothbrush on days 5 and 7. Based on these results, we cannot rule out the possibility that increased FN production due to electric toothbrush vibration contributes to the maturation of the ECM in gingival tissue. Therefore, further studies based on this hypothesis are warranted. The mRNA expression of col I was upregulated on days 3, 5, and 7 of culture following the vibration compared with that in the control. However, the protein level of col I increased only on day 3 following the vibration compared with that in the control. No difference was found in the control between days 5 and 7. The effects of the vibration of an electric toothbrush on the protein expression of col I were not in accordance with the effect on mRNA expression. Clark et al. [35] reported that when the required amount of collagen matrix is synthesized, TGF-β produced by fibroblasts acts on the fibroblasts themselves to regulate collagen matrix synthesis. Similar to these findings, in the present study, col I expression increased at the mRNA level on day 5. On the contrary, the protein expression of col I was not increased even when the stimulation time was increased because the required amount of collagen was already synthesized in the stimulation group on day 3 and its expression was regulated.

Previous studies have reported that physical and mechanical forces induce the production of the ECM through the FAK-ERK pathway in fibroblasts [36,37]. Therefore, we examined FAK signal transduction in fibroblasts stimulated with the vibration of an electric toothbrush. The results revealed that FAK phosphorylation was activated 15 min after the stimulation with an electric toothbrush and the activation continued for up to 1 h. Next, the cells were treated with siRNA of PTK2. As expected, FAK expression was attenuated in PTK2 siRNA-treated cells. Moreover, PTK2 siRNA diminished the upregulatory effects of vibration on the expression of col I, col III and elastin. These results suggested that vibratory stimulation with an electric toothbrush increased col III and elastin expression via FAK signal transduction. In contrast, elevated FN expression in cells stimulated with vibration was not blocked by PTK2 siRNA.

It has been reported that applying orthodontic mechanical forces on HGF results in a partial increase in cell proliferation and collagen gene expression via the TGF-β signaling pathway [38,39]. Row et al. [40] reported that cadherin-11 regulates collagen and elastin expression in fibroblasts via the TGF-β and ROCK pathways. From these reports, it can be deduced that the stimulation with the electric toothbrush may have increased elastin production through not only FAK but also TGF-β.

In our study, the expression of col I, col III, and elastin and synthesis of FN in HGnFs increased following electric toothbrush stimulation. In addition, we demonstrated that these effects were likely mediated by the FAK signaling pathway. Our findings suggest that mechanical stimulation with an electric toothbrush promotes ECM protein expression via FAK signaling and other pathways.

Almeida et al. [41] reported on collagen expression in healthy gingival tissue and diseased gingival tissue. According to this report, the expression of col I is higher in healthy gingiva than in diseased gingival tissues, but there is no difference in the expression of col III. Collagen plays an important role in the healing of connective tissue by providing tissue strength and a scaffold for cell adhesion and migration. During the phase of new tissue formation, the ECM is poorly organized [42]. When periodontal tissue is regressed, owing to periodontal disease or aging, col III deposition is considerably increased, supporting healing. Particularly in gingival tissue, during the healing phase, col III is the dominant component initially but is resorbed over time and replaced by col I fibers [43]. In the present study, col III expression increased immediately after stimulation with an electric toothbrush, suggesting that the use of an electric toothbrush may promote the repair of connective tissue damaged by periodontal disease and aging.

Although there are many epidemiological studies on electric toothbrushes, there are no studies that have elucidated the mechanisms by which electric toothbrushes affect fibroblasts in in vitro studies. We believe that this study may have explored a new aspect of electric toothbrushes. On the other hand, it is essential to verify whether the results of the present study occur in the human oral cavity, which is considered a limitation of this study.

## 5. Conclusions

We investigated the effect of exposure to vibratory stimulation with an electric toothbrush on ECM protein production in gingival fibroblasts under the conditions of a healthy periodontium. Our findings suggested that electric toothbrush stimulation increases the expression of col I, col III, elastin, and FN via the FAK signaling pathway in HGnF cells. However, this study was conducted in vitro, and in vivo studies will be essential in the future.

## Figures and Tables

**Figure 1 biomolecules-12-00771-f001:**
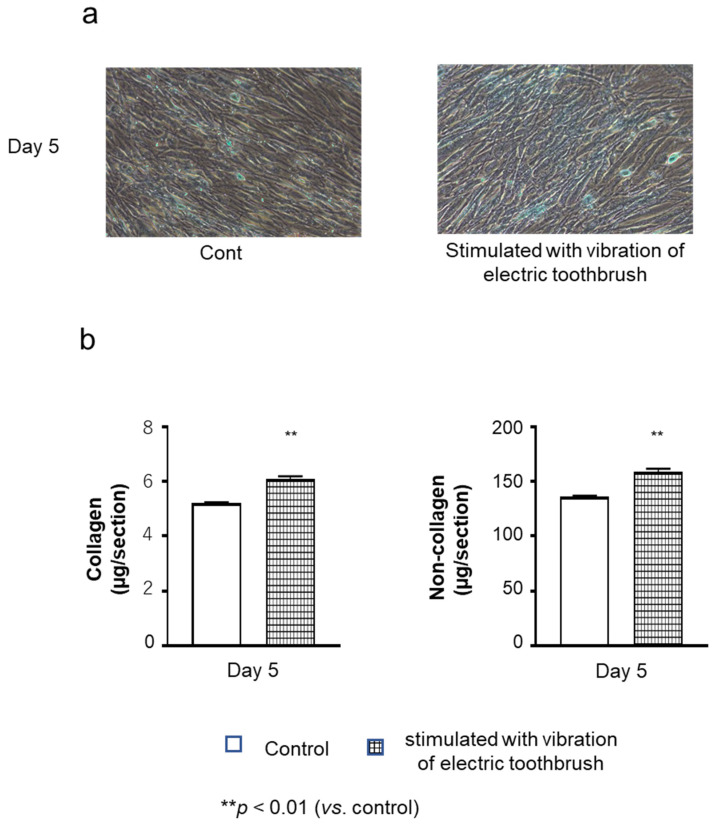
Effects of exposure to vibratory stimulation with an electric toothbrush on the amount of collagenous and non-collagenous proteins in cultured cell layers. HGnFs were exposed to vibratory stimulation with an electric toothbrush for 5 days. Collagenous (red) and non-collagenous (green) proteins were examined by Sirius red and Fast green staining on day 5 of culture (**a**), and quantitative analysis (**b**) was conducted using the stain extraction buffer. Each bar indicates mean ± standard deviation (SD) of five independent experiments. ** *p* < 0.01 (vs. control).

**Figure 2 biomolecules-12-00771-f002:**
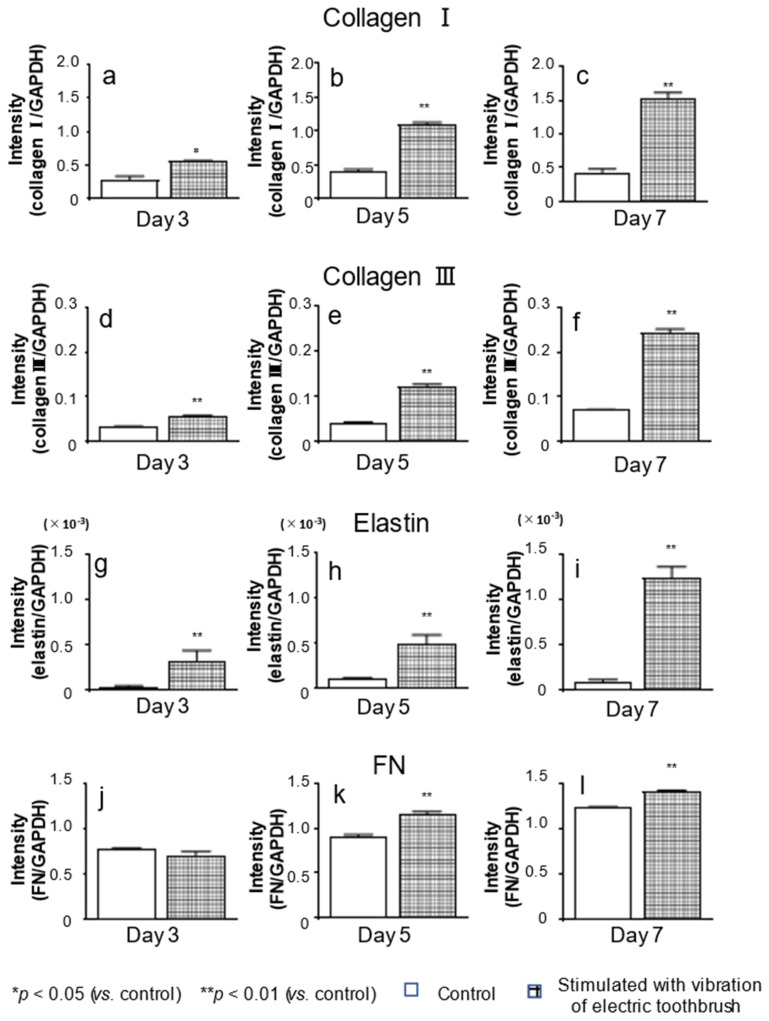
Effects of exposure to vibratory stimulation with an electric toothbrush on the mRNA expression of ECM factors in gingival fibroblasts. HGnFs were exposed to vibratory stimulation with an electric toothbrush for 7 days. The mRNA expression of ECM factors (**a**–**l**) was quantified using real-time PCR. Each bar indicates mean ± standard deviation (SD) of five independent experiments. * *p* < 0.05, ** *p* < 0.01 (vs. control).

**Figure 3 biomolecules-12-00771-f003:**
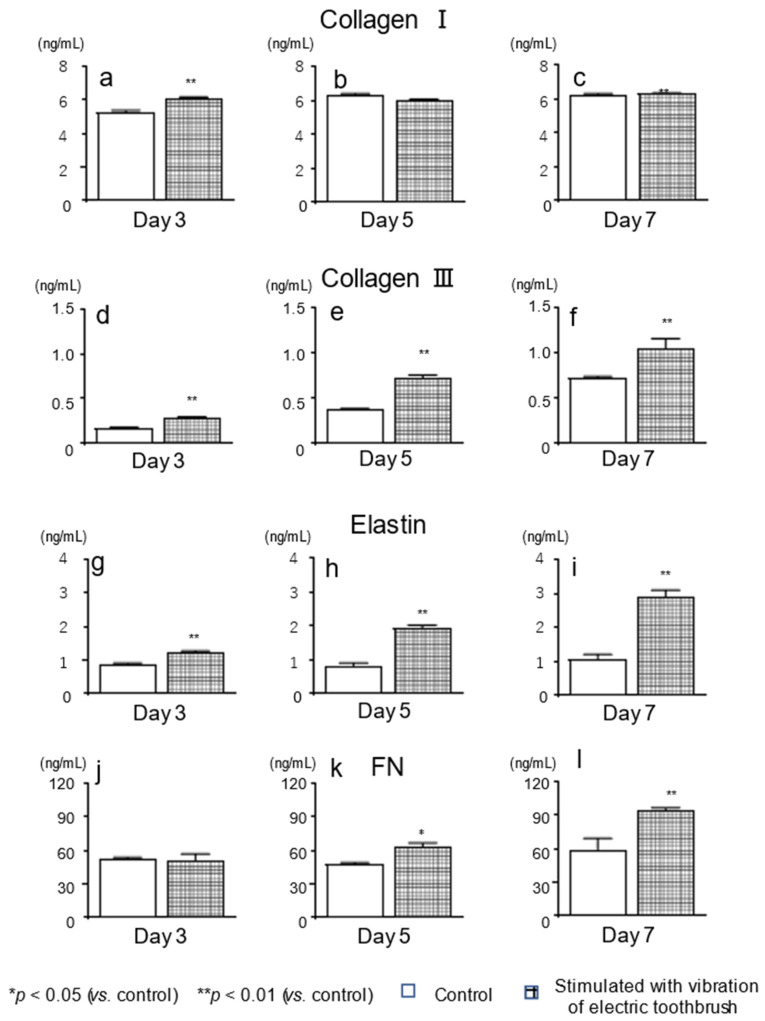
Effects of exposure to vibratory stimulation with an electric toothbrush on protein expression of ECM factors in gingival fibroblasts. HGnFs were exposed to vibratory stimulation with an electric toothbrush for 7 days. The protein expression of four ECM factors (**a**–**l**) in the culture supernatants and lysate was quantified using ELISA. Each bar indicates mean ± standard deviation (SD) of five independent experiments. * *p* < 0.05, ** *p* < 0.01 (vs. control).

**Figure 4 biomolecules-12-00771-f004:**
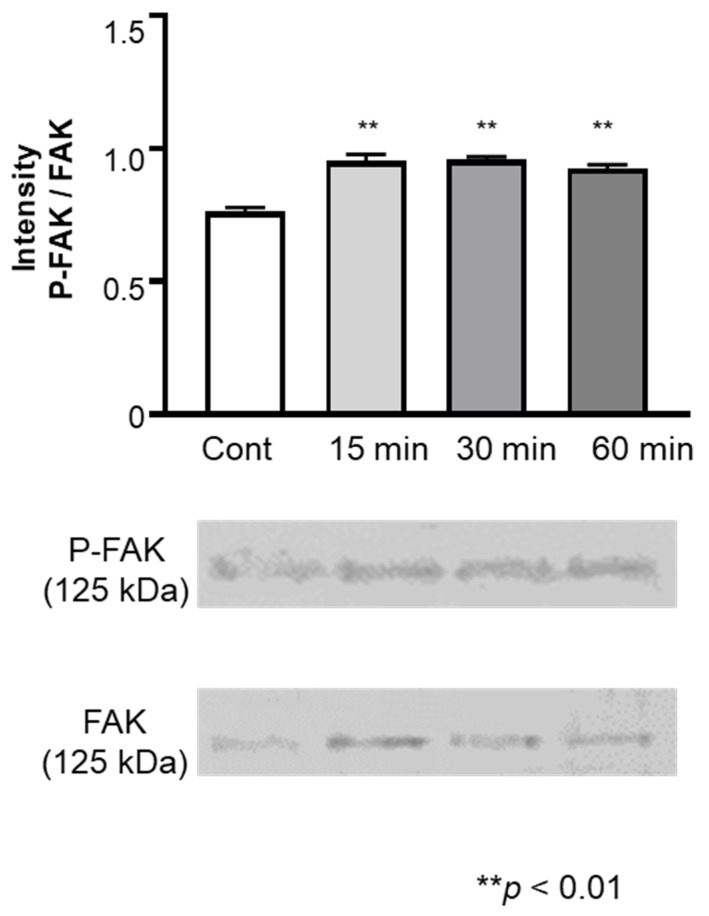
Effect of exposure to vibratory stimulation with an electric toothbrush on the expression of FAK and phosphorylated FAK in gingival fibroblasts. HGnFs were exposed to vibratory stimulation with an electric toothbrush for 15, 30, and 60 min. Each bar indicates mean ± standard deviation (SD) of three independent experiments. ** *p* < 0.01.

**Figure 5 biomolecules-12-00771-f005:**
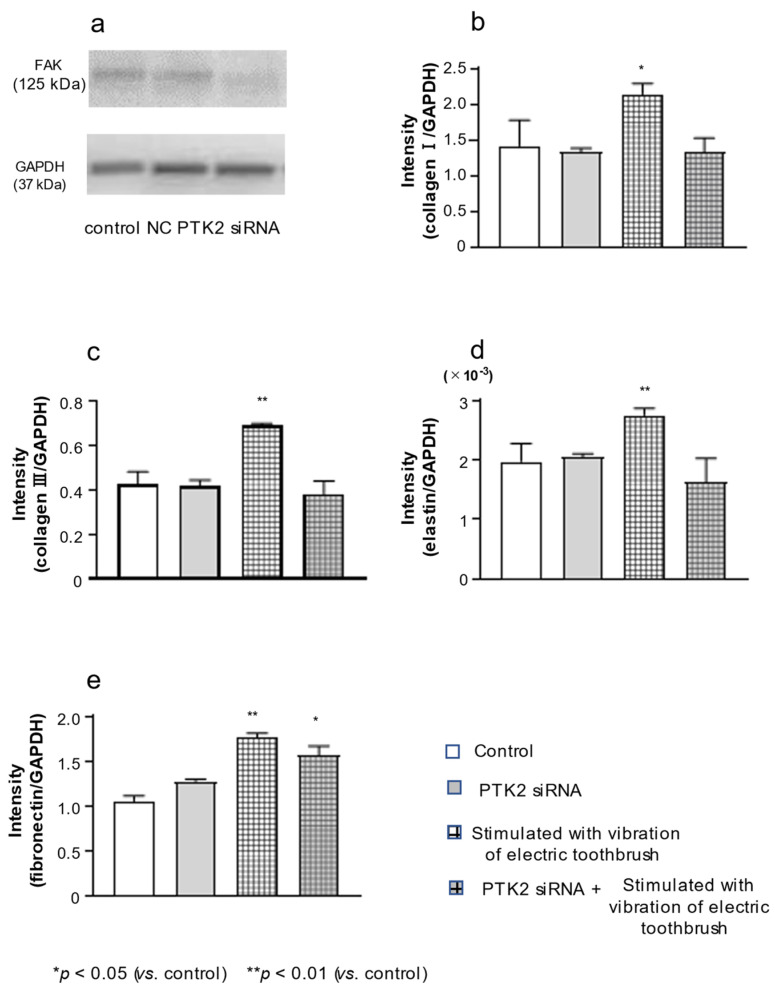
Effect of PTK2 siRNA on the expression of FAK, type III collagen, elastin, and fibronectin in gingival fibroblasts. HGnFs were transfected with PTK2 siRNA or its negative control (NC) (**a**). The mRNA expression of col III, elastin, and FN was quantified 5 days after treatment with PTK2 siRNA (**b**–**e**). Each bar indicates mean ± standard deviation (SD) of five independent experiments. * *p* < 0.05, ** *p* < 0.01.

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
