# Peer review of "Effects of Electric-Toothbrush Vibrations on the Expression of Collagen and Non-Collagen Proteins through the Focal Adhesion Kinase Signaling Pathway in Gingival Fibroblasts"

_biomolecules, 2022, doi:10.3390/biom12060771_

Round 1

Reviewer 1 Report

Thanks for submitting this interesting manuscript. 

I am recommending to include some recently published literature on relevant applications, mainly in the field of orthodontic treatment and tooth repositioning, which you definitely should include in your introduction as well as especially in your discussion. Those include, but are not necessarily limited to Changkhaokan et al 2022 in Angle Orthod, mouthpiece applications as reported by Akbari et al 2021 in Am J Orthod Dentofacial Orthop, papers by Tarek El-Bialy and Thomas Shipley (e. g. Dent J 2020, J Orthod Sci 2019, Dent J 2018). 

Having said that, you have proven the principle of effects of vibration under use of a side-to-side toothbrush technology with approx. 270 Hz. You quoted other power toothbrush technologies with other frequencies as well. Given this fact and the above mentioned mouthpiece technologies, I think you may want to consider testing the effects of such devices in your model as well. At least you should acknowledge this in your manuscript. 

In that context and given the approach with a limited range of devices and of exposure, I highly recommend to add a chapter outlining the strengths and also the limitations of your study approach. 

Specific comments on the manuscript:

Line 52: The author of reference 10 is Dr. Ccahuana-Vasquez, so pls quote by family name.

Line 106ff: It seems important that you mention and standardize the pressure of application of the toothbrush to the bottom of your plate, since there will be an effect of the degrees of freedom of the brush head and maybe even activated pressure control, which may slow down the motor/mechanical activity. You may want to elaborate on how you controlled for pressure and applied the brush. Thus, the 31000 strokes/min may actually be different resulting in a different frequency as well (line 110). I do not understand the comment on the frequency of 261 Hz and "recommended by the manufacturer". How can they recommend a technical feature inherent to their technology (line 112)?

Lines 252-273: The description of proteins  is too long and if considered critical should be moved into the Introduction section. In the discussion the authors should focus on their results.

I recommend to remove speculative wording (lines 248-251, and 274-276) and address that as recommended above in limitations of the study in terms of what cannot be concluded for sure but may warrant further investigation based on such hypotheses.

Lines 287/288: I would recommend a statement that this is in vitro research and in vivo relevance is to be proven.

Lines 296-298: Provide evidence for the consideration.

Line 315: Typo: TGF-ß. And in addition, I again recommend to be more careful with the wording as long as you do not have an experimental verification.

Lines 321/322: I think "diseased" is missing at the beginning of Line 322.

Line 331: change to "toothbrush may promote the repair....".

Author Response

I am recommending to include some recently published literature on relevant applications, mainly in the field of orthodontic treatment and tooth repositioning, which you definitely should include in your introduction as well as especially in your discussion. Those include, but are not necessarily limited to Changkhaokan et al 2022 in Angle Orthod, mouthpiece applications as reported by Akbari et al 2021 in Am J Orthod Dentofacial Orthop, papers by Tarek El-Bialy and Thomas Shipley (e. g. Dent J 2020, J Orthod Sci 2019, Dent J 2018). 

Response: We thank you for the comment. We have cited the suggested studies in the manuscript.

Having said that, you have proven the principle of effects of vibration under use of a side-to-side toothbrush technology with approx. 270 Hz. You quoted other power toothbrush technologies with other frequencies as well. Given this fact and the above mentioned mouthpiece technologies, I think you may want to consider testing the effects of such devices in your model as well. At least you should acknowledge this in your manuscript. 

Response: We thank you for the comment. We have added the necessary information and made necessary revisions in the manuscript (line 67).

In that context and given the approach with a limited range of devices and of exposure, I highly recommend to add a chapter outlining the strengths and also the limitations of your study approach. 

Response: We thank you for the comment. We have added the necessary information and made necessary revisions in the manuscript.

Specific comments on the manuscript:

Line 52: The author of reference 10 is Dr. Ccahuana-Vasquez, so pls quote by family name.

Response: We thank you for pointing this out. We have corrected it.

Line 106ff: It seems important that you mention and standardize the pressure of application of the toothbrush to the bottom of your plate, since there will be an effect of the degrees of freedom of the brush head and maybe even activated pressure control, which may slow down the motor/mechanical activity. You may want to elaborate on how you controlled for pressure and applied the brush. Thus, the 31000 strokes/min may actually be different resulting in a different frequency as well (line 110). I do not understand the comment on the frequency of 261 Hz and "recommended by the manufacturer". How can they recommend a technical feature inherent to their technology (line 112)?

Response: We thank you for the comment. We have made the necessary revision in the manuscript.line109-116

Lines 252-273: The description of proteins is too long and if considered critical should be moved into the Introduction section. In the discussion the authors should focus on their results.

Response: We thank you for the comment. We have made the necessary revision in the manuscript.

I recommend to remove speculative wording (lines 248-251, and 274-276) and address that as recommended above in limitations of the study in terms of what cannot be concluded for sure but may warrant further investigation based on such hypotheses.

Response: We thank you for the comment. We have made the necessary revision in the manuscript.

Lines 287/288: I would recommend a statement that this is in vitro research and in vivo relevance is to be proven.

Response: We thank you for the comment. We have made the necessary revision in the conclusions.

Lines 296-298: Provide evidence for the consideration.

Response: We thank you for the comment. We have made the necessary revision in the manuscript.

Line 315: Typo: TGF-ß. And in addition, I again recommend to be more careful with the wording as long as you do not have an experimental verification.Lines 321/322: I think "diseased" is missing at the beginning of Line 322.Line 331: change to "toothbrush may promote the repair....".

Response: We thank you for the comment. We have made the necessary revision in the manuscript.

Reviewer 2 Report

Title: Effects of electric toothbrush vibrations on the expression of collagen and non-collagen proteins through the focal adhesion kinase signaling pathway in gingival fibroblasts.

Manuscript ID: biomolecules-1748393

The article is interesting, it highlights a very little known effect of electric toothbrushes on gingival tissues. It is well written. Just some issues that could improve it, are mentioned below.

-Keywords. Use appropriate keywords so that if the article is published, it can be properly located in the bibliographic repertoires.

-Introduction. Could the authors document, in two, three lines with the corresponding reference, if the clinical effectiveness of current electric toothbrushes is superior to manual ones? Although it is not the objective of the study, for readers it would increase its interest and importance.

-Results. Could histological images be improved? The color of sirius red stains, for example, are not appreciated.

-Discussion. Could the authors make any consideration and/or applicability of their results, on the periodontal ligament, a structure very rich in collagen, elastin fibers and at the same time highly vascularized?

-Conclusion, lines 334 -338. These conclusions do not reflect the results of the study. It is not possible to refer to periodontal disease when the fibroblasts did not come from patients with periodontitis nor were they stimulated with LPS from periodontopathogens. The appropriate conclusions in the opinion of this reviewer are those expressed in the penultimate paragraph, lines 316-320.

Author Response

The article is interesting, it highlights a very little known effect of electric toothbrushes on gingival tissues. It is well written. Just some issues that could improve it, are mentioned below.

-Keywords. Use appropriate keywords so that if the article is published, it can be properly located in the bibliographic repertoires.

Response: We thank you for the comment. We have revised the Keywords.

-Introduction. Could the authors document, in two, three lines with the corresponding reference, if the clinical effectiveness of current electric toothbrushes is superior to manual ones? Although it is not the objective of the study, for readers it would increase its interest and importance.

Response: We thank you for the comment. We have added the necessary information in the manuscript (line 54).

-Results. Could histological images be improved? The color of sirius red stains, for example, are not appreciated.

Response: We thank you for the comment. We have made the necessary revision in the manuscript.

-Discussion. Could the authors make any consideration and/or applicability of their results, on the periodontal ligament, a structure very rich in collagen, elastin fibers and at the same time highly vascularized?

Response: In the present study, we assumed that electric toothbrush stimulation affects the connective tissue just below the epithelium. (The bottom of the plate and the epithelial tissue are approximately 0.02 inch thick). Therefore, if it can be proven that electric toothbrush stimulation can reach the periodontal ligament, the present study results could be applied in the periodontal ligament as well.

-Conclusion, lines 334 -338. These conclusions do not reflect the results of the study. It is not possible to refer to periodontal disease when the fibroblasts did not come from patients with periodontitis nor were they stimulated with LPS from periodontopathogens. The appropriate conclusions in the opinion of this reviewer are those expressed in the penultimate paragraph, lines 316-320.

Response: We have made the necessary revision in the manuscript.

Reviewer 3 Report

The article “Effects of electric toothbrush vibrations on the expression of 2 collagen and non-collagen proteins through the focal adhesion 3 kinase signaling pathway in gingival fibroblasts”, is very interesting and I have some suggestions that I believe could improve the manuscript.

In abstract you should clarify that col III is type III collagen

Line 63 You should also clarify that the study from Kusano et at. was performed in dogs “The effect of electric toothbrush vibrations on collagen production in connective tissue has been studied in vivo and in vitro [14,15].

Line 52: Renzo et al. [10], it's more appropriate Ccahuana-Vasquez et al

Line 312: Sindhu et al. [35], it is Row et al

Line 321: Tatiane et al. [36], it is Almeida et al

The authors should point out the limitations of the study. For example, 1) it is an in vitro study; 2) control is over unstimulated cells….

When the authors confirm “Our findings suggested that electric toothbrush stimulation exerts beneficial effects on the healing of wounds caused by periodontal disease and aging”, this conclusion cannot be subtracted from the results of this study, further research would be needed.

Author Response

The article “Effects of electric toothbrush vibrations on the expression of 2 collagen and non-collagen proteins through the focal adhesion 3 kinase signaling pathway in gingival fibroblasts”, is very interesting and I have some suggestions that I believe could improve the manuscript.

In abstract you should clarify that col III is type III collagen

Response: We thank you for the comment. We have made the necessary revision in the manuscript.

Line 63 You should also clarify that the study from Kusano et at. was performed in dogs “The effect of electric toothbrush vibrations on collagen production in connective tissue has been studied in vivo and in vitro [14,15].

Response: We thank you for the comment. We have made the necessary revision in the manuscript.

Line 52: Renzo et al. [10], it's more appropriate Ccahuana-Vasquez et al

Line 312: Sindhu et al. [35], it is Row et al

Line 321: Tatiane et al. [36], it is Almeida et al

Response: We thank you for the comment. We have made the necessary revisions in the manuscript.

The authors should point out the limitations of the study. For example, 1) it is an in vitro study; 2) control is over unstimulated cells….

When the authors confirm “Our findings suggested that electric toothbrush stimulation exerts beneficial effects on the healing of wounds caused by periodontal disease and aging”, this conclusion cannot be subtracted from the results of this study, further research would be needed.

Response: We thank you for the comment. We have revised the conclusion accordingly.